## PERSPECTIVE

# Gas channels and CO₂ transport across cell membranes: Mechanisms and synergy

Nazih L. Nakhoul 

*Department of Medicine, Section of Nephrology Tulane Medical School, New Orleans, Louisiana, USA*

Email: nakhoul@tulane.edu

The peer review history is available in the Supporting Information section of this article (https://doi.org/10.1113/JP290205#support-information-section).

### CO₂ transport across cell membranes

The transport of gases, such as $CO_2$, $NH_3$ and $O_2$, across cell membranes is critical for all forms of life. It was generally accepted that gases freely and passively cross the cell membrane barrier governed by the well-established solubility-diffusion model. As early as 1830 it was established that *solubility* across membranes increases with the lipophilic nature of the molecule. Shortly thereafter the principle of *'diffusion'* was introduced to the solubility concept. In 1855 Fick's theory of diffusion established the parameters that govern the flow of solutes. Extending the solubility-diffusion model to include membrane barriers, the rate of transport of $CO_2$ across a cell membrane can be expressed as $J_{CO2} = P_{CO2} \cdot ([CO_2]o - [CO_2]i)$, where $J_{CO2}$ is the flux of $CO_2$ and $P_{CO2}$ is the permeability of $CO_2$ driven by the concentration gradient of $CO_2$ across the membrane. $P_{CO2}$ includes the diffusion coefficient of $CO_2$ ($D_{CO2}$) and the area of $CO_2$ exposure and is inversely related to the barrier thickness (bulk solution and cell membrane).

Around 1900 Meyer and Overton, working independently on anaesthetics, recognized an association between the lipid solubility of substances and their increased ability to cross the membrane. This association, known as the Meyer–Overton rule, is inaccurately often used to mean the solubility-diffusion model.

### Control of CO₂ diffusion across cellular membranes

Building on the principles of diffusion, a cell can influence membrane transport of $CO_2$ by controlling *membrane permeability* or the *$CO_2$ concentration gradient* across the cell membrane. Membrane permeability can be affected by altering the composition of the lipid bilayer or by the presence of membrane proteins. Abundance of the hydrophilic membrane proteins, which do not conduct $CO_2$, and their arrangement can usually hinder or obstruct $CO_2$ access to the membrane.

On the other hand, maintaining the $CO_2$ concentration gradient is dependent on $CO_2$ build-up in unstirred layers adjacent to the membrane. When a cell is exposed to $CO_2$-containing solution, it diffuses through the extracellular unstirred layer, through the membrane and then through an intracellular layer before moving to the intracellular compartment. Accumulation of $CO_2$ in the membrane intracellular unstirred layer will reduce the concentration gradient and slow diffusion of $CO_2$. Converting intracellular $CO_2$ to $HCO_3^-$ will keep low intracellular $CO_2$ concentration adjacent to the membrane. Similarly $CO_2$ in the extracellular unstirred layer needs to be replenished from $HCO_3^-$ to maintain the gradient. Interconversion of $CO_2$ and $HCO_3^-$ is a slow process that is highly accelerated by carbonic anhydrases (CAs).

### Gas channels and CO₂ transport

The widespread acceptance of Overton's rule that all gases cross all cell membranes by dissolving in the lipid layer of the cell membrane was challenged by two important discoveries. Boron's group in 1994 (Waisbren et al., 1994) reported that specific cell membranes were relatively impermeable to $CO_2$. This was followed by another seminal discovery that the water channel AQP1, expressed in oocytes, increased the permeability of the membrane to $CO_2$, thus acting as a $CO_2$ channel (Nakhoul et al., 1998). Using pH microelectrodes to measure intracellular pH, exposure of the oocyte to a solution containing $CO_2/HCO_3^-$ caused intracellular acidification due to the intracellular reaction: $CO_2 + H_2O \leftrightarrow HCO_3^- + H^+$. As such $CO_2$ transport was estimated from the $pH_i$ decrease and the rate of intracellular acidification. Because hydration of $CO_2$ is slow, build-up of $CO_2$ in the intracellular unstirred layer could decrease the concentration gradient of $CO_2$. Indeed injecting the oocyte with CA increased $CO_2$ influx significantly in oocytes expressing AQP1 compared to $H_2O$-injected oocytes.

### Synergistic mechanisms

This narrative briefly describes the important hallmarks of studying $CO_2$ transport across cell membranes. The important points include (i) the permeability and diffusion of $CO_2$, (ii) the role of CAs and (iii) the AQPs as $CO_2$ channels. Boron's group has contributed significantly to elucidating the mechanisms of all these points. In those studies identifying cell membranes that are impermeable to $CO_2$ led to the discovery of AQP1 as a $CO_2$ channel. In a series of several elegant papers, they examined the role of CA in affecting $CO_2$ transport. In the first paper (Musa-Aziz et al., 2014a) they addressed the role of intracellular CA. They showed that injecting oocytes with soluble CA-II accelerated $CO_2$ influx and this influx was blocked by ethoxzolamide. In the second paper (Musa-Aziz et al., 2014b) they examined the effect of extracellular CA by expressing CA-IV at the cell membrane. The same group then used mathematical modelling to interpret the $CO_2$ influx in the oocyte.

The paper in this issue by Wang et al. (2025) examines all three factors that were previously studied independently to address $CO_2$ membrane transport. They systematically assessed functional interactions among AQP5 (a water channel expressed in the lung), intracellular CA and extracellular CA. They used demanding simultaneous measurements of intracellular and membrane-surface pH to measure $CO_2$ transport in presence and absence of AQP5. This approach made it possible to determine the response in $CO_2$ transport when changes and measurements were made on the same side of the membrane (e.g. intracellular CA-II and intracellular pH) named 'cis measurements' or on opposite sides of the membrane (e.g. CA-II and surface pH) named 'trans measurements'. The results of the study demonstrated independent cooperation among intracellular and extracellular CA and AQP5, which does not involve direct physical interaction but augmented $CO_2$ transport to more than an additive effect

The Journal of Physiology

in what is termed 'synergistic' effect. These findings are novel and provide important insights into the factors that govern $CO_2$ transport across cell membranes.

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

## Additional information

### Competing interests

No competing interests declared.

### Author contributions

N.L.N.: conception or design of the work; drafting the work or revising it critically for important intellectual content; final approval of the version to be published; agreement to be accountable for all aspects of the work.

### Funding

No funding.

### Keywords

AQP5, carbonic anhydrase II, carbonic anhydrase IV

### Supporting information

Additional supporting information can be found online in the Supporting Information section at the end of the HTML view of the article. Supporting information files available:

**Peer Review History**

