## [Peer Review History · The Journal of Physiology]

Gas Channels and CO₂ Transport Across Cell Membranes: Mechanisms and Synergy

Nazih L. Nakhoul
DOI: 10.1113/JP290205

Corresponding author(s): Nazih Nakhoul (nakhoul@tulane.edu)

The following individual(s) involved in review of this submission have agreed to reveal their identity: Walter F Boron (Referee #1)

Review Timeline:

Submission Date:	07-Nov-2025
Editorial Decision:	19-Nov-2025
Revision Received:	24-Nov-2025
Editorial Decision:	26-Nov-2025
Revision Received:	03-Dec-2025
Accepted:	05-Dec-2025

Senior Editor: Peking Fong

Reviewing Editor: Peking Fong

Transaction Report:

Dear Dr Nakhoul,

Re: JP-P-2025-290205 "**Gas Channels and CO₂ Transport Across Cell Membranes: Mechanisms and Synergy**" by Nazih L. Nakhoul

Thank you for submitting your manuscript to The Journal of Physiology. It has been assessed by a Reviewing Editor and by 1 expert referee and we are pleased to tell you that it is acceptable for publication following satisfactory revision.

The review comments are copied at the end of this email.

Please address all the points raised and incorporate all requested revisions or explain in your Response to Referees why a change has not been made. We hope you will find the comments helpful and that you will be able to return your revised manuscript within 2 weeks. If you require longer than this, please contact journal staff: jp@physoc.org.

REVISION CHECKLIST:

We look forward to receiving your revised submission.

Yours sincerely,

Peying Fong
Senior Editor
The Journal of Physiology

REQUIRED ITEMS

- 1) - The reference list must be in alphabetical order, rather than numbered, to comply with our Journal format.

- 2) - Please include a full, separate title page as part of your main article (Word) file, which should contain the following: title, authors, affiliations, corresponding author name and contact details, keywords, and running title.

- 3) - The corresponding author must provide an institutional email address in the manuscript.

EDITOR COMMENTS

Senior Editor:

Your Perspectives article has been reviewed. Please refer to the marked-up version of the manuscript that is attached.

REFEREE COMMENTS

Referee #1:

Dr Boron sent a direct email to Dr Nakhoul with additional comments.

END OF COMMENTS

The Journal of Physiology

<https://jp.msubmit.net>

JP-P-2025-290205

**Title: Gas Channels and CO₂ Transport Across Cell Membranes:
Mechanisms and Synergy**

Authors: Nazih Nakhoul

Author Conflict: No competing interests declared

Author Contribution: Nazih Nakhoul: Conception or design of the work;
Drafting the work or revising it critically for important intellectual content;
Final approval of the version to be published; Agreement to be accountable
for all aspects of the work

Running Title:

Dual Publication: N/a

Funding: No Funding: Nazih L. Nakhoul, No Funding; No Funding: Nazih
L. Nakhoul, No Funding This is an invited perspective. No funding is needed

Disclaimer: This is a confidential document.

1 Gas Channels and CO₂ Transport Across Cell Membranes: Mechanisms and Synergy

2 CO₂ transport across cell membranes

3 The transport of gases, such as CO₂, NH₃ and O₂, across cell membranes is critical for all forms
4 of life. It was generally accepted that gases freely and passively cross the cell membrane barrier
5 governed by the well-established solubility-diffusion model known as the Meyer-Overton rule.
6 According to this rule, the rate of movement of gasses across cell membranes depends on both
7 solubility and diffusion of the gas. As early as 1830, Mitchel established that solubility across
8 membranes increases with the lipophilic nature of the molecule. In 1855, Fick's theory of
9 diffusion established the parameters that govern flow of solutes across the membrane barriers.
10 Applying the solubility-diffusion model, the rate of transport of CO₂ across a cell membrane can
11 be expressed as: $J_{CO_2} = P_{CO_2} \cdot ([CO_2]_o - [CO_2]_i)$

12 where J_{CO_2} is flux of CO₂, P_{CO_2} is permeability of CO₂ driven by the concentration gradient of
13 CO₂ across the membrane. P_{CO_2} includes the diffusion coefficient of CO₂ (D_{CO_2}), the area of CO₂
14 exposure and inversely related to the barrier thickness (bulk solution and cell membrane).

15 Control of CO₂ diffusion across cellular membranes

16 Building on the principles of diffusion, a cell can influence membrane transport of CO₂ by
17 controlling **membrane permeability** or the **CO₂ concentration gradient** across the cell
18 membrane. Membrane permeability can be affected by altering the composition of the lipid
19 bilayer or by the presence of membrane proteins. In mammalian cells, cholesterol is a main
20 component of cell membranes and raising its content was shown to reduce CO₂ permeability.
21 Abundance of the hydrophilic membrane proteins, that do not conduct CO₂, and their
22 arrangement can usually hinder or obstruct CO₂ access to the membrane.

23 On the other hand, maintaining the CO₂ concentration gradient is dependent on CO₂ buildup in
24 unstirred layers adjacent to the membrane that will alter the concentration gradient. When a cell
25 is exposed to CO₂-containing solution, it diffuses through the extracellular unstirred layer,
26 diffuses through the membrane and then through an intracellular layer before moving to the
27 intracellular compartment. Accumulation of CO₂ in the membrane intracellular unstirred layer will
28 reduce the concentration gradient and slow diffusion of CO₂. Converting intracellular CO₂ to
29 HCO₃⁻ will keep low intracellular CO₂ concentration adjacent to the membrane. Similarly, CO₂ in
30 the extracellular unstirred layer ~~need~~ to be replenished from HCO₃⁻ to maintain the gradient.
31 Interconversion of CO₂ and HCO₃⁻ is a slow process that is highly accelerated by carbonic
32 anhydrases (CA).

33 Gas channels and CO₂ transport

34 The widespread acceptance of Overton's rule that all gases cross all cell membranes by
35 dissolving in the lipid layer of the cell membrane was challenged by two important discoveries.
36 Boron's group in 1994 (1) reported that specific cell membranes were relatively impermeable to
37 CO₂. This was followed by another seminal discovery that the water channel AQP1, expressed
38 in oocytes, increased permeability of the membrane to CO₂ thus acting as a CO₂ channel (2).

39 Using pH microelectrodes to measure intracellular pH, exposing the oocyte to a solution
40 containing $\text{CO}_2/\text{HCO}_3^-$ caused intracellular acidification due to the intracellular reaction:

41 $\text{CO}_2 + \text{H}_2\text{O} \rightleftharpoons \text{HCO}_3^- + \text{H}^+$. As such, CO_2 transport was estimated from the pH_i decrease and
42 the rate of intracellular acidification. Because hydration of CO_2 is slow, buildup of CO_2 in the
43 intracellular unstirred layer could decrease the concentration gradient of CO_2 . Indeed, injecting
44 the oocyte with CA increased CO_2 influx significantly in oocytes expressing AQP1 compared to
45 H_2O -injected oocytes.

46 Synergistic Mechanisms

47 The above narrative briefly describes important hallmarks of studying CO_2 transport across cell
48 membranes. The important points include: i) Permeability and diffusion of CO_2 , ii) the role of
49 carbonic anhydrases, and iii) AQPs as CO_2 channels. Boron's group has contributed
50 significantly to elucidating the mechanisms of all these points. In those studies, identifying cell
51 membranes that are impermeable to CO_2 led to discovery of AQP1 as a CO_2 channel, as
52 described above. In a series of several elegant papers, they examined the role of CA in affecting
53 CO_2 transport. In the first paper (3), they addressed the role of intracellular CA by injecting the
54 oocytes with soluble CA-II. They showed that CA-II accelerated CO_2 influx and was blocked by
55 ethoxzolamide. In the second paper (4), they examined the effect of extracellular CA by
56 expressing CA-IV at the cell membrane. In the third paper (5), they used sophisticated
57 mathematical modeling to interpret the CO_2 influx in the oocyte.

58 The paper in this issue by Wang, Moss and Boron examines all three factors that were
59 previously studied independently to address CO_2 membrane transport. They systematically
60 assessed functional interactions among AQP5 (a water channel expressed in the lung),
61 intracellular CA and extracellular CA. They used demanding simultaneous measurements of
62 intracellular and membrane-surface pH to measure CO_2 transport in the presence and absence
63 of AQP5. This approach made it possible to determine the response in CO_2 transport when
64 changes and measurements were made on the same side of the membrane (e.g. intracellular
65 CA-II and intracellular pH) named "cis measurements" or on opposite sides of the membrane
66 (e.g. CA-II and surface pH) named "trans measurements". The results of the study
67 demonstrated independent cooperation among intracellular and extracellular CA and AQP5, that
68 does not involve direct physical interaction to promote CO_2 transport in what they termed
69 "synergy" effect. These findings are novel and provide new insights into studying the factors that
70 affect CO_2 transport across cell membranes.

- 71 1. **Waisbren SJ, Geibel J, Boron WF, and Modlin IM.** Luminal perfusion of
72 isolated gastric glands. *Am J Physiol* 266: C1013-1027, 1994.
- 73 2. **Nakhoul NL, Davis BA, Romero MF, and Boron WF.** Effect of expressing the
74 water channel aquaporin-1 on the CO_2 permeability of *Xenopus* oocytes [see
75 comments]. *Am J Physiol* 274: C543-548, 1998.
- 76 3. **Musa-Aziz R, Occhipinti R, and Boron WF.** Evidence from simultaneous
77 intracellular- and surface-pH transients that carbonic anhydrase II enhances CO_2 fluxes
78 across *Xenopus* oocyte plasma membranes. *Am J Physiol Cell Physiol* 307: C791-813,
79 2014.

- 80 4. **Musa-Aziz R, Occhipinti R, and Boron WF.** Evidence from simultaneous
81 intracellular- and surface-pH transients that carbonic anhydrase IV enhances CO₂
82 fluxes across *Xenopus* oocyte plasma membranes. *Am J Physiol Cell Physiol* 307:
83 C814-840, 2014.
- 84 5. **Occhipinti R, Musa-Aziz R, and Boron WF.** Evidence from mathematical
85 modeling that carbonic anhydrase II and IV enhance CO₂ fluxes across *Xenopus*
86 oocyte plasma membranes. *Am J Physiol Cell Physiol* 307: C841-858, 2014.
- 87

Response to referee

I thank the referee for the comments on the historical chronology related to the development of the solubility-diffusion model. Hopefully the contributions and dates are now clear.

The term “synergy” is now better defined to mean an augmented effect that is more than a strictly additive effect.

Dear Dr Nakhoul,

Re: JP-P-2025-290205R1 "**Gas Channels and CO₂ Transport Across Cell Membranes: Mechanisms and Synergy**" by Nazih L. Nakhoul

Thank you for submitting your manuscript to The Journal of Physiology. It has been assessed by a Reviewing Editor and by 0 expert referee and we are pleased to tell you that it is acceptable for publication following satisfactory revision.

The review comments are copied at the end of this email.

Please address all the points raised and incorporate all requested revisions or explain in your Response to Referees why a change has not been made. We hope you will find the comments helpful and that you will be able to return your revised manuscript within 2 weeks. If you require longer than this, please contact journal staff: jp@physoc.org.

REVISION CHECKLIST:

We look forward to receiving your revised submission.

Yours sincerely,

Peying Fong
Senior Editor
The Journal of Physiology

EDITOR COMMENTS

Senior Editor:

Review of your Perspectives manuscript, "Gas Channels and CO₂ Transport Across Cell Membranes: Mechanisms and Synergy" is now complete.

Please note that while this revised version now satisfactorily addresses points raised by the Referee during initial review, there appear to be several lapses. Please review the guidelines pertaining to Perspectives articles that appear within Information for Authors at

https://jp.msubmit.net/cgi-bin/main.plex?form_type=display_requirements#perspectivesguidelines.

Specifically, although all references included in the References section indeed now are ordered according to The Journal of Physiology's style (alphabetically, not numbered), many appear missing from this list. These include not only those historical references appearing in the first section (Mitchell, 1830; Graham, 1854; Graham, 1866; Meyer, 1899; Overton, 1901), but also the reference to the paper being highlighted (Wang et al, 2025). Including all of these would bring the total number of references well above the limit of 5, specifically to 11.

We look forward to receiving your revised manuscript and thank you for your contributions to The Journal of Physiology.

END OF COMMENTS

Answer to reviewer's comments:

I thank the Senior Editor for the comment that “many references appear missing from this list. These include not only those historical references appearing in the first section (Mitchell, 1830; Graham, 1854; Graham, 1866; Meyer, 1899; Overton, 1901), but also the reference to the paper being highlighted (Wang et al, 2025). Including all of these would bring the total number of references well above the limit of 5, specifically to 11”.

Response:

I am aware that the limit on the references is 5. Because of this limitation, I removed direct reference to the published manuscripts but kept the historical timeline to highlight the history related to this work. I hope that this will do.

The second point is that I do not have the exact full reference to the current Walter's manuscript (Wang et al.' 2025) since it is not yet published in J Physiol. I included it in the Bibliography, but it may need editorial correction.

I hope that you find these corrections adequate.

Dear Dr Nakhoul,

Re: JP-P-2025-290205R2 "**Gas Channels and CO₂ Transport Across Cell Membranes: Mechanisms and Synergy**" by Nazih L. Nakhoul

We are pleased to tell you that your paper has been accepted for publication in The Journal of Physiology.

Please note that Perspective articles are not typically covered by institutional open access agreements with our publisher, Wiley. Wiley do not offer article processing charge (APC) discounts for smaller article types in hybrid subscription journals, meaning that if you wish for your Perspective to be published Open Access, you will have to pay the full APC. As such, we recommend authors publish Perspectives 'behind the paywall', where they will become freely accessible after a 12-month embargo (i.e. please select the NON open access option via Wiley Author services during proofing).

Should you wish to pay for Open Access, you will be able to place an order by logging into Wiley Author services.

Yours sincerely,

Peying Fong
Senior Editor
The Journal of Physiology

IMPORTANT POINTS TO NOTE FOLLOWING ACCEPTANCE OF YOUR PAPER:

- **IMPORTANT NOTICE ABOUT OPEN ACCESS:** To assist authors whose funding agencies mandate immediate public access to published research findings, The Journal of Physiology allows authors to pay an Open Access (OA) fee to have their papers made freely available immediately on publication.

- You can help your research get the attention it deserves! Check out Wiley's free Promotion Guide for best-practice recommendations for promoting your work at: www.wileyauthors.com/eoo/guide. You can learn more about Wiley Editing Services which offers professional video, design, and writing services to create shareable video abstracts, infographics, conference posters, lay summaries, and research news stories for your research at: www.wileyauthors.com/eoo/promotion.

- If you would like to receive our 'Research Roundup', a monthly newsletter highlighting the cutting-edge research published in The Physiological Society's family of journals (The Journal of Physiology, Experimental Physiology, Physiological Reports, The Journal of Nutritional Physiology and The Journal of Precision Medicine: Health and Disease), please click this link, fill in your name and email address and select 'Research Roundup':

<https://www.physoc.org/journals-and-media/membernews>

EDITOR COMMENTS

Reviewing Editor:

REFEREE COMMENTS